# Image-to-image translation for cross-domain disentanglement

**Abel Gonzalez-Garcia**
Computer Vision Center
agonzalez@cvc.uab.es

**Joost van de Weijer**
Computer Vision Center
Universitat Autònoma de Barcelona

**Yoshua Bengio**
MILA
Université de Montréal

## Abstract

Deep image translation methods have recently shown excellent results, outputting high-quality images covering multiple modes of the data distribution. There has also been increased interest in disentangling the internal representations learned by deep methods to further improve their performance and achieve a finer control. In this paper, we bridge these two objectives and introduce the concept of cross-domain disentanglement. We aim to separate the internal representation into three parts. The shared part contains information for both domains. The exclusive parts, on the other hand, contain only factors of variation that are particular to each domain. We achieve this through bidirectional image translation based on Generative Adversarial Networks and cross-domain autoencoders, a novel network component. Our model offers multiple advantages. We can output diverse samples covering multiple modes of the distributions of both domains, perform domain-specific image transfer and interpolation, and cross-domain retrieval without the need of labeled data, only paired images. We compare our model to the state-of-the-art in multi-modal image translation and achieve better results for translation on challenging datasets as well as for cross-domain retrieval on realistic datasets.

## 1 Introduction

Deep learning has greatly improved the quality of image-to-image translation methods. These methods aim to learn a mapping that transforms images from one domain to another. Examples include colorization, where the aim is to map a grayscale image to a plausible colored image of the same scene [21, 47], and semantic segmentation, where an RGB image is translated to a map indicating the semantic class of each pixel in the RGB image [13, 31]. Isola et al. [23] proposed a general purpose image-to-image translation method. Their method is successfully applied to a wide range of problems when paired data is available. The theory is further extended to unpaired data by introducing a cycle consistency loss [49]. The U-Net [40] architecture is commonly used for image-to-image translation. This network can be interpreted as an encoder-decoder network. The encoder extracts the relevant information from the input domain and passes it on to the decoder, which then transforms this information to the output domain. In spite of the current popularity of these models, the learned representation (output of the encoder) has only been studied for some related tasks [32, 44]. Here we investigate and impose structure to the specific representation learned in image-to-image translation models.

Disentangling the accidental scene events, such as illumination, shadows, viewpoint and object orientation from the intrinsic scene properties has been a long desired goal of computer vision [6, 42]. When applied to deep learning, this allows deep models to be aware of isolated factors of variation affecting the represented entities [7, 34]. Therefore, models can marginalize information along a particular factor of variation, should it be not relevant for the task at hand. Such a process can be especially beneficial for tasks that are hindered by the presence of particular factors, for example, varying illumination conditions in object recognition. Moreover, disentangled representations grant a more precise control for those tasks that perform actions based on the representation.

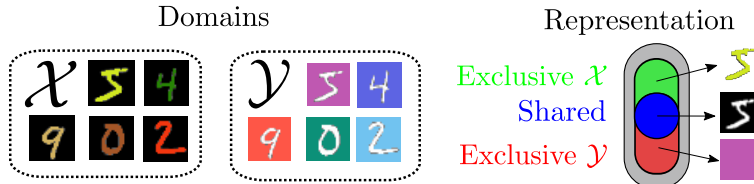

Figure 1: (Left) Example of a pair of domains containing images with colored digits on black background or white digits on colored background. (Right) Disentangled representation, separated into shared part across domains (digit) and domain-exclusive parts (color in the background or on the digit).

In this paper, we combine the disentanglement objective with image-to-image translation, and introduce the concept of *cross-domain disentanglement*. The aim is to disentangle the domain specific factors from the factors that are shared across the domains. To do so, we partition the representation into three parts; the *shared* part containing information that is common to both domains, and two *exclusive* parts, which only represent those factors of variation that are particular to each domain (see example in figure 1). Cross-domain disentanglement for image-to-image translation has several advantages that allow for applications that would otherwise not be feasible: (i) *Sample diversity:* we can generate a distribution of images conditioned on the input image, whereas most image-to-image architectures can only generate deterministic results [23, 49]. Our approach is similar to the recent work of Zhu et al. [50], although we explicitly model variations in both domains, whereas they only consider variations in the output domain; (ii) *Cross domain retrieval:* we can retrieve similar images in both domains based on the part of the representation that is shared between the domains, and, unlike [4], we do not require labeled data to learn the shared representation; (iii) *Domain-specific image transfer:* domain-specific features can be transferred between images; and (iv) *Domain-specific interpolation:* we can interpolate between two images with respect to domain-specific features.

Our model is based on bidirectional image translation across domains, using a pair of Generative Adversarial Networks (GANs) [17]. We enforce a disentangled structure in the learned representation through an adequate combination of multiple losses and a new network component called cross-domain autoencoder. We demonstrate the disentanglement properties of our method on variations on the MNIST dataset [26], and apply it to bidirectional multi-modal image translation in more complex datasets [3, 38], achieving better results than state-of-the-art methods [23, 50] due to the finer control and generality granted by our disentangled representation. Additionally, we outperform [50] in cross-domain retrieval on realistic datasets [23, 45]. Our code and models are publicly available at https://github.com/agonzgarc/cross-domain-disen.

## 2   Cross-domain disentanglement networks

The goal of our method is to learn deep structured representations that are clearly separated in three parts. Let $\mathcal{X}, \mathcal{Y}$ be two image domains (e.g. fig. 1) and let $R$ be an image representation in either domain. We split $R$ into sub-representations depending on whether the information contained in that part belongs exclusively to domain $\mathcal{X}$ ($E^{\mathcal{X}}$), domain $\mathcal{Y}$ ($E^{\mathcal{Y}}$), or it is shared between both domains ($S^{\mathcal{X}}/S^{\mathcal{Y}}$). Figure 1 depicts an example of this representation for images of digits with colors in different areas (digit or background). In this case, the shared part of the representation is the actual digit without color information, i.e. "the image contains a 5". The exclusive parts are the color information in the different parts of the image, e.g. "the digit is yellow" or "the background is purple".

Figure 2 presents an overview of our model, which can be separated into image translation modules (left) and cross-domain autoencoders (right). The translation modules $G$ and $F$ translate images from domain $\mathcal{X}$ to domain $\mathcal{Y}$, and from $\mathcal{Y}$ to $\mathcal{X}$, respectively. They follow an encoder-decoder architecture. Encoders $G_e$ and $F_e$ process the input image through a series of convolutional layers and output a latent representation $R$. Traditionally in these architectures (e.g. [23, 49, 50]), the decoder takes the full representation $R$ and generates an image in the corresponding output domain. In our model, however, the latent representation is split into shared and exclusive parts, i.e. $R = (S, E)$, and only the shared part of the representation is used for translation. Decoders $G_d$ and $F_d$ combine $S$ with random noise $z$ that accounts for the missing exclusive part, which is unknown for the other domain at test time. This enables the generation of multiple plausible translations given an input image. The other component of the model, the cross-domain autoencoders, is a new type of module that helps aligning the latent distributions and enforce representation disentanglement. The following sections describe all the components of the model and detail how we achieve the necessary constraints on the learned representation. For simplicity, we focus on input domain $\mathcal{X}$, the model for $\mathcal{Y}$ is analogous.

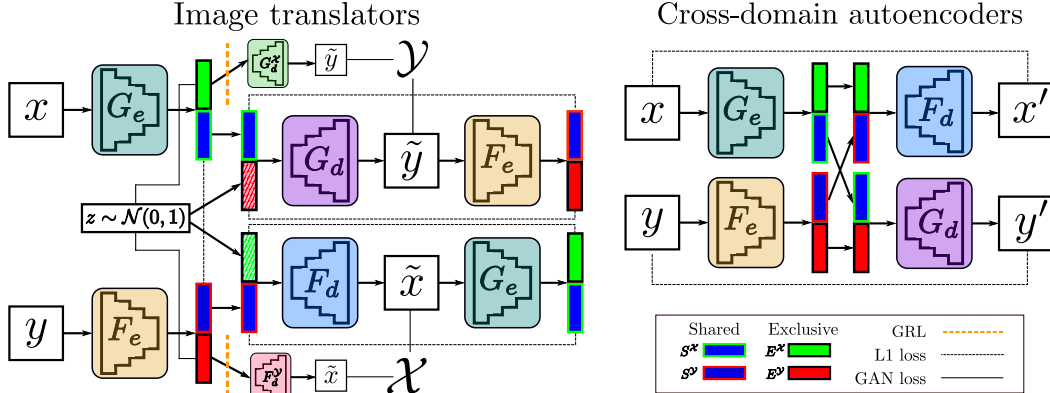

Figure 2: Overview of our model. (Left) Image translation blocks, $G$ and $F$, based on an encoder-decoder architecture. We enforce representation disentanglement through the combination of several losses and a GRL. (Right) Cross-domain autoencoders help aligning the latent space and impose further disentanglement constraints.

## 2.1 Image translation modules

Generative Adversarial Networks (GAN) are a popular framework [17] consisting of two networks that compete against each other. The *generator* tries to synthesize realistic images to fool a *discriminator*, whose task is to detect whether images come from the generator or from the real data distribution. When the generated images are conditioned using an input image, the task becomes image translation. Our image translation modules are inspired by the successful architecture used in pix2pix [23], based on convolutional GANs [36]. The generator encoder consists of several convolutional layers of stride 2, followed by batch normalization [22] and leaky ReLU activations. The decoder uses fractionally strided convolutions to upsample the internal representation back to the image resolution. We adapt this architecture for the disentanglement problem with the following modifications.

**Exclusive representation.** The exclusive representation $E^{\mathcal{X}}$ of an image $x \in \mathcal{X}$ must not contain information about domain $\mathcal{Y}$. Therefore, it should not be possible to use only $E^{\mathcal{X}}$ to generate an image in $\mathcal{Y}$. To enforce this desirable behavior, we try to generate $\mathcal{Y}$ images from $E^{\mathcal{X}}$ but also actively guide the feature learning to prevent this from happening. For this, we propose a novel application of the *Gradient Reversal Layer* (GRL), originally introduced in [16] to learn domain-agnostic features. During the forward pass of the network, this layer acts as the identity function. On the backward pass, however, the GRL *reverses* the gradients flowing back from the corresponding branch. Inspired by this idea, our model includes a small decoder in each image translation module, $G_d^{\mathcal{X}}$ in the $G$ case, that tries to generate images in $\mathcal{Y}$ with $E^{\mathcal{X}}$ as input. We add a GRL at the beginning of $G_d^{\mathcal{X}}$, immediately after $E^{\mathcal{X}}$ (orange dashed line in fig. 2). The GRL inverts the sign of the gradient that is backpropagated to the encoder $G_e$, affecting only those units involved in the generation of the exclusive features $E^{\mathcal{X}}$. We train $G_d^{\mathcal{X}}$ with an adversarial loss on the generated images. In theory, this approach will force $E^{\mathcal{X}}$ not to contain information that might generate images in the $\mathcal{Y}$ domain.

**Shared representation.** The shared parts of the representations $S^{\mathcal{X}}$ and $S^{\mathcal{Y}}$ of a pair of corresponding images $(x, y) \in (\mathcal{X}, \mathcal{Y})$ should contain similar information and be invariant to the domain. Some domain adaptation approaches [16, 8, 9] have successfully used a GRL to create domain-invariant features. However, we have found here that this approach can quickly become unstable when the loss starts diverging. A possible solution for this consists in bounding the loss by the performance of random chance [14]. In our case, and due to the fact that our images are paired, we can attain the desired invariance simply by adding an L1 loss on these features, which forces them to be indistinguishable for both domains:

$$\mathcal{L}_S = \mathbb{E}_{x \sim \mathcal{X}, y \sim \mathcal{Y}} \left[ \left\| S^{\mathcal{X}} - S^{\mathcal{Y}} \right\| \right]. \tag{1}$$

**Adding noise in the representation.** A drawback of loss (1) on the shared representation is that it encourages the model to use a small signal $\left\| S^{\mathcal{X}} \right\| \to 0$, which reduces the loss but does not increase the similarity between $S^{\mathcal{X}}$ and $S^{\mathcal{Y}}$. We have found out that adding small noise ($\mathcal{N}(0, 0.1)$) to the output of the encoder as in [46] prevents this from happening and leads to better results[1].

**Architectural bottleneck.** Most image translation approaches [5, 23, 33, 49, 50] are devised for pairs of domains that retain the spatial structure (e.g. grayscale to color images), and thus a great amount of information is shared between input and output. In the usual encoder-decoder architecture, all this information passes through a bottleneck, here called the latent representation, that connects the two components. To prevent the loss of details at higher resolutions, it is common to use skip connections (e.g. U-Net [5, 23, 33, 40, 49, 50]). When disentangling the latent representation, however, skip connections pose a problem. The higher resolution features operated by the encoder contain both shared and exclusive information, but the decoder must receive only the shared part of the representation from the encoder. Therefore, instead of using skip connections, we reduce the architectural bottleneck by increasing the size of the latent representation. In fact, we only increase the spatial dimensions of the shared part of the representation, from $1 \times 1 \times 512$ to $8 \times 8 \times 512$. We found out that in the considered domains, the exclusive part can be successfully modeled by a $1 \times 1 \times 8$ vector, which is later tiled and concatenated with the shared part before decoding. We implement the different size of the latent representation by parallel last layers in the encoder, convolutional for the shared part and fully connected for the exclusive part.

**Reconstructing the latent space.** The input of the translation decoders is the shared representation $S$ and random input noise that takes the role of the exclusive part of the representation. Concretely, we use an 8-dimensional noise vector $z$ sampled from $\mathcal{N}(0, I)$. The exclusive representation must be approximately distributed like the input noise, as both take the same place in the input of the decoder (see sec. 2.2). To achieve this, we add a discriminator $D_z$ that tries to distinguish between the output exclusive representation $E^{\mathcal{X}}$ and input noise $z$, and train it with the original GAN loss [17]. This pushes the distribution of $E^{\mathcal{X}}$ towards $\mathcal{N}(0, I)$ and makes the input of the decoder consistent.

Commonly, adversarial image translation approaches [23, 49, 50] attempt to achieve some stochasticity by adding random noise to the input or the internal features. However, in many cases this noise is ignored and the generated outputs are uni-modal [23, 49]. We follow an idea explored in [11, 12, 50] to avoid this and reconstruct the latent representation from the generated image by feeding it to the encoder. The reconstructed representation should match the decoder input, so we add an L1 loss between the original and reconstructed $S^{\mathcal{X}}$, as well as the input noise $z$ and the reconstructed exclusive part

$$\mathcal{L}_{\text{recon}}^{\mathcal{X}} = \mathbb{E}_{x \sim \mathcal{X}}\big[||G_e(G_d(S^{\mathcal{X}}, z)) - (S^{\mathcal{X}}, z)||\big]. \tag{2}$$

**WGAN-GP loss.** Since the original formulation [17], more advanced GAN losses have appeared. For example, the use of the Wasserstein-1 distance in WGAN [2] has been shown to provide desirable convergence properties and to correlate well with perceptual quality of the generated images. We adopt the more stable Gradient Penalty variant [18] (WGAN-GP) for our model. Let $D$ be a convolutional discriminator with a single scalar as output. Following [23], we condition $D$ on the corresponding paired image in the input domain, which is concatenated to the real or generated image (omitted from following notation). Our discriminator and generator losses are then defined as

$$\mathcal{L}_{\text{Disc}}^{\mathcal{X}} = \mathbb{E}_{\tilde{x} \sim \widetilde{\mathcal{X}}}[D(\tilde{x})] - \mathbb{E}_{x \sim \mathcal{X}}[D(x)] + \lambda \cdot \mathbb{E}_{\hat{x} \sim \widehat{\mathcal{X}}}[(||\nabla_{\hat{x}} D(\hat{x})||_2 - 1)^2], \tag{3}$$

$$\mathcal{L}_{\text{Gen}}^{\mathcal{X}} = -\mathbb{E}_{\tilde{x} \sim \widetilde{\mathcal{X}}}[D(\tilde{x})], \tag{4}$$

where $\widehat{\mathcal{X}}$ is the distribution obtained by randomly interpolating between real images $x$ and generated images $\tilde{x}$ [18]. Contrarily to [23], we do not include a reconstruction term $||\tilde{x} - x||$ in the generator loss. Our outputs should cover multiple modes of the output distribution and thus they do not necessarily match the paired image in the other domain. We combine both GAN losses in $\mathcal{L}_{\text{GAN}}^{\mathcal{X}}$.

## 2.2 Cross-domain autoencoders

The image translation modules impose three main constraints: (1) the shared part of the representation must be identical for both domains, (2) the exclusive part only has information about its own domain, and (3) the generated output must belong to the other domain. However, there is no force that aligns the generated output with the corresponding input image to show the same concept (e.g. same number) but in different domains. In fact, the generated images need not correspond to the input if the encoders learn to map different concepts to the same shared latent representation. In order to achieve consistency across domains, we introduce the idea of cross-domain autoencoders (fig. 2, right).

A classic autoencoder would take the full representation encoded for input image $x$, $G_e(x) = (S^{\mathcal{X}}, E^{\mathcal{X}})$ and input it in the decoder of module $F$, which outputs images in $\mathcal{X}$, with the goal of reconstructing $x$. Since the shared representations in our model must be indistinguishable, we could

use the shared representation $S^{\mathcal{Y}}$ from the other domain instead of $S^{\mathcal{X}}$. This provides an extra incentive for the encoder to place useful information about domain $\mathcal{X}$ in $E^{\mathcal{X}}$, as $S^{\mathcal{Y}}$ does not contain any domain-exclusive information. Our cross-domain autoencoders use this combination to generate the reconstructed input $x' = F_d(S^{\mathcal{Y}}, E^{\mathcal{X}})$. We train them with the standard L1 reconstruction loss

$$\mathcal{L}_{\text{auto}}^{\mathcal{X}} = \mathbb{E}_{x \sim \mathcal{X}} \left[ ||x' - x|| \right]. \tag{5}$$

### 2.3 Bi-directional image translation

Given the multi-modal nature of our system in both domains, our architecture is unified to perform image translation in the two directions simultaneously. This is paramount to learn how to disentangle what part of the representation can be shared across domains and what parts are exclusive to each. We train our model jointly in an end-to-end manner, minimizing the following total loss

$$\begin{aligned}
\mathcal{L} = &w_{\text{GAN}}(\mathcal{L}_{\text{GAN}}^{\mathcal{X}} + \mathcal{L}_{\text{GAN}}^{\mathcal{Y}}) + w_{\text{Ex}}(\mathcal{L}_{\text{GAN}}^{G_d^{\mathcal{X}}} + \mathcal{L}_{\text{GAN}}^{F_d^{\mathcal{Y}}}) \\
&+ w_{\text{L1}}(\mathcal{L}_S + \mathcal{L}_{\text{auto}}^{\mathcal{X}} + \mathcal{L}_{\text{auto}}^{\mathcal{Y}} + \mathcal{L}_{\text{recon}}^{\mathcal{X}} + \mathcal{L}_{\text{recon}}^{\mathcal{Y}}).
\end{aligned} \tag{6}$$

## 3 Related work

**Disentangling deep representations.** A desirable property of learned representations is the ability to disentangle the factors of variation [7]. For this reason, there has been a substantial interest on learning disentangled representations [19, 43], including some work based on generative models [25, 34, 37]. One of the earliest architectures for learning disentangled representations using deep learning was applied to the task of emotion recognition [39]. The work of [34] combines a Variational Autoencoder (VAE) with a GAN to disentangle representations depending on what is specified (i.e. labeled in the dataset) and the remaining factors of variation, which are unspecified. In a similar intra-domain spirit, InfoGAN [10] optimizes a lower bound on the mutual information between the representation and the images, successfully controlling some factors of variation in the considered images. Reed et al. [37] propose learning each factor of variation of the image manifold as its own sub-manifold using a higher-order Boltzmann machine. Alternatively, the analogy-making approach of [38] attempts to disentangle the factors of variation by using representation arithmetic. Finally, some domain adaptation approaches [8, 9, 16, 30] aim at obtaining invariant features for the classification task, granting some level of disentanglement but depending on class labels. Even though representation disentangling has been widely studied, we are unaware of any work studying true cross-domain representation disentangling, which is the focus of this paper.

**Image translation.** Lately, image generation using adversarial training methods has attracted a great amount of attention [2, 17]. We consider the *image translation* task, in which the generative process is conditioned on an input image [23, 47, 50]. While some approaches had previously applied adversarial losses for specific image translation tasks such as style transfer [28] or colorization [47], the approach of Isola et al. [23], called pix2pix, was the first GAN-based image translation approach that was not tailored to a specific application. Despite the excellent results of these models, they are limited by the lack of variation of their generated outputs, which are virtually deterministic, as the input noise is mostly ignored. As a consequence, they can only provide a one-to-one mapping across image domains, a phenomenon named mode collapse [41].

In order to reduce this limitation, Zhu et al. [50] extended the pix2pix framework. They minimize the reconstruction error of the latent code by a reverse decoder with the generated output as input, forcing the generator to take the input noise into account. Furthermore, they combine a conditional GAN with a conditional VAE, whose goal is to provide a plausible latent vector given a target image. The resulting model, called BicyleGAN, effectively achieves one-to-many image translations. However, there are several differences with our method. Theirs is restricted in only one direction, whereas our method operates in a many-to-many setting. Moreover, our representation grants a finer control on the stochastic factors of the generated images as we also model variations on the inputs, allowing us to keep selected properties fixed. Finally, the obtained disentangled image features are useful for additional tasks beyond image translation, such as cross-domain retrieval or visual analogies.

Concurrently to our work, several approaches [1, 20, 27, 29] have attempted to improve on image-to-image translation by disentangling the internal representation into *content* and *style*, which enable the effective generation of multi-modal outputs. In a similar spirit, Ma et al. [32] disentangle the features learned for person image generation into foreground, background, and pose. In our case, however, image translation is not the only final task: we demonstrate the generality of our distentangled features by applying them to other tasks such as cross-domain retrieval or domain-specific image transfer.

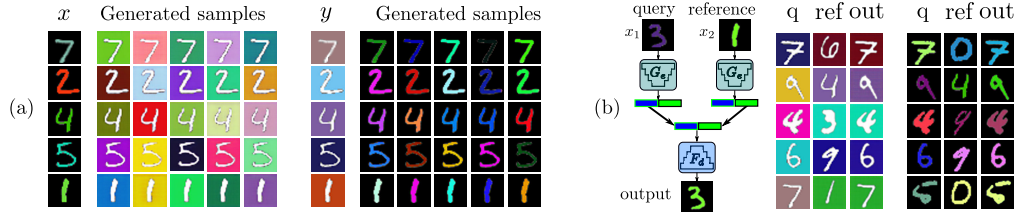

Figure 3: (a) Samples generated by our model using random noise as exclusive representation, where $\mathcal{X}$ = MNIST-CD and $\mathcal{Y}$ = MNIST-CB. (b) Visual analogies, combining shared and exclusive parts of different samples.

## 4 Experiments

### 4.1 Representation disentangling on MNIST variations

We evaluate the properties of our representation by following the protocol introduced by [34] to measure the disentanglement of a representation. However, [34] operates within one domain only, whereas we learn cross-domain representations. For this reason, we extend MNIST [26], the handwritten-digit dataset used in [34], with variations that correspond to two different domains. In our variations, we either colorize the digit (MNIST-CD) or the background (MNIST-CB) with a randomly chosen color (fig.1). We use the standard splits for train (50K images) and test (10K images). We detail the architectures and hyperparameters used for all experiments in the supplementary material.

**Sample diversity.** Figure 3a shows samples generated by our model in both domains using random input noise. We can observe how our model successfully generates diverse samples for different noise values, varying the color where appropriate but maintaining the digit information. Note, however, that the model has no knowledge of the digit in the image as labels are not provided, it effectively learns what information is shared across both domains. This demonstrates that we achieve many-to-many image translation through proper manipulation of the disentangled latent representation.

**Domain-specific image transfer.** We evaluate our domain-specific transferring capabilities using *visual analogy generation*, which consists in applying a particular property of a given reference image to a query image [38] (e.g. changing the digit color in one MNIST-CD image to the another image's color, as in fig. 3b). Our disentangled representation grants us a precise control over the image generation process, facilitating the visual analogy task as it can be seen as applying domain-specific properties (encoded in the exclusive part of the representation) from one image to another. Our model can generate visual analogies as follows. Let us consider two input images from one domain $x_1, x_2 \in \mathcal{X}$ and their disentangled representations $R_1 = (S_1^{\mathcal{X}}, E_1^{\mathcal{X}})$, and $R_2 = (S_2^{\mathcal{X}}, E_2^{\mathcal{X}})$, respectively. We use $x_1$ as input query and $x_2$ as reference. We can generate the desired visual analogy by simply combining the shared part of the query with the exclusive part of the reference and running it through the decoder, i.e. $F_d(S_1^{\mathcal{X}}, E_2^{\mathcal{X}})$. Fig. 3b illustrates this process and shows qualitative results. The query images acquire the corresponding properties of the reference images, as the output images have the correct digit and color. Note how our model has not been explicitly trained to achieve this behavior, it is a natural consequence of a correctly disentangled representation.

**Domain-specific image interpolation.** Beyond transferring domain-specific properties between images, our representation allows us to interpolate between two images along domain-specific properties. To do this, we simply keep one part of the representation fixed while we interpolate between two samples in the other part. Finally, we combine each interpolated value with the fixed part and run it through the decoder to generate the image. Figure 4a shows results for interpolations on the exclusive and shared parts for various random samples of both domains. When interpolating on the exclusive part, we generate samples along domain-specific factors of variation, i.e. color, and maintain the digit, which is the shared information. Analogously, when we interpolate on the shared part, the domain-specific properties stay stable while one digit smoothly transforms into the other.

**Cross-domain retrieval.** Given an image query and an image database, the goal of retrieval is selecting those images that are similar to the query, either semantically or visually. In *cross-domain* retrieval [4, 24, 35], query and database are from different domains. Generally, the shared information between domains is semantic whereas the exclusive is stylistic, and so our disentangled representation enables both semantic and visual retrieval using either of its parts. We perform cross-domain retrieval using Euclidean distance between shared features, and compare it with a simple baseline using distances on image pixels. Table 1 (left) presents the results in terms of the common retrieval metric of Recall@1. The high values obtained by shared features show how using our representation provides an effective approach for cross-domain retrieval, clearly superior to directly using image pixels. Moreover, we do not need image labels, as opposed to specialized approaches such as [4].

Table 1: Cross-domain retrieval (Recall@1) on MNIST-CD/CB and Facades [45]/Maps [23].

| Method | MNIST-CD → CB | MNIST-CB → CD | Method | Facades | | Maps | |
|---|---|---|---|---|---|---|---|
| | | | | F → L | L → F | S → M | M → S |
| Pixels | 30.45 | 40.02 | BicycleGAN | - | 45 | - | 68.0 |
| Shared | 99.99 | 99.95 | Ours | 95 | 97 | 91.4 | 96.9 |
| Exclusive | 9.93 | 9.80 | | | | | |

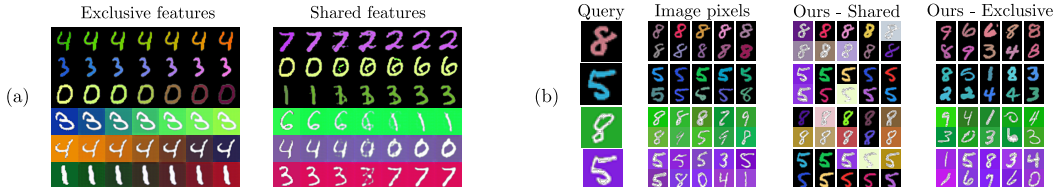

Figure 4: (a) Interpolation between samples on either part of the representation. (b) Random image queries and their 10 nearest-neighbors in the set union of both domains, using distance on pixels or parts of our representation.

To further demonstrate representation disentanglement we perform retrieval from one domain to a database of images from both domains. Figure 4b displays examples of random queries in this setting, showing the 10 nearest-neighbors using distances on image pixels, shared, and exclusive features. Shared features retrieve images of the same digit from both domains (46%), and thus contain scarce domain-specific information, whereas using pixels clearly prioritizes images from the query domain (almost 100%). Moreover, exclusive features retrieve images that are visually similar regardless of the digit, indicating that our exclusive features may be used for purely visual retrieval.

## 4.2 Many-to-many image translation

In this section, we demonstrate the performance of our method for the task of bi-directional multi-modal image translation. We use pix2pix [23] and the state-of-the-art multi-modal approach of BicycleGAN [50] as baselines, combining two independently trained models in either direction. Despite the truly remarkable results of these approaches, they are limited by the underlying assumption of spatial correspondence between images across domains, and thus cannot be applied when domains undergo significant structural changes such as viewpoint, as confirmed experimentally. Our method, on the other hand, removes this assumption as it does not rely on additional side information to generate its samples, only on the learned latent representation. To provide a fair comparison, we remove the skip connections in [23, 50] and increment the latent space to $8 \times 8$, as in our architecture. This architectural adjustment increases the model's ability to translate images with significant structural changes, such as those used in this section. For completeness, we also provide quantitative results for the original BicycleGAN in Table 2. We measure the performance quantitatively with the Learned Perceptual Image Patch Similarly (LPIPS) metric of [48], which is based on differences between network features and correlates very well with human judgments. We use the official implementation and default settings by the authors [48].

**3D car models.** We use the 3D car images [38] of the 199 CAD models in [15], rendered from 24 equally spaced viewpoints. Let $\mathcal{X}$ be the frontal/rear car views, and $\mathcal{Y}$ the profile views. We set 5 random cars for test and train with the remaining 796 images. Fig. 5 shows generated samples by our method and the baselines. The deterministic nature of pix2pix conflates both views into one, making it unable to output realistic cars with a specific viewpoint. BicycleGAN generates better samples, but the quality is still rather poor. Our method generates good quality samples and covers multiple modes of the output distribution. Moreover, it maintains shared information across domains (e.g. car color), whereas BicycleGAN's samples might not correspond to the input image. We attribute this to our finer control over the latent representation, as we model image variations also in the input domain.

Table 2 presents quantitative results. For each image in the test set (two views per car model/domain) we generate three samples and compute the LPIPS [48] metric between them and both possible ground-truths as domains are bi-modal, e.g. left and right profiles. Then, we select the minimum distance to either ground-truth and average over samples. In both directions, our model outperforms the two baselines, generating samples that are perceptually more similar to the actual examples.

Fig. 5b shows visual analogies for cars (created as in fig. 3b). With our exclusive representation, we can apply the orientation of one car to another while maintaining other properties such as style or color. Finally, we show in fig. 5c samples of our model when domain $\mathcal{X}$ is the frontal view and $\mathcal{Y}$ contains all other views. Even for this more challenging case, we manage to output samples covering

Table 2: LPIPS metric on samples generated for cars [38] and chairs [3].

| Method | Cars | | Chairs | |
|---|---|---|---|---|
| | $\mathcal{X} \to \mathcal{Y}$ | $\mathcal{Y} \to \mathcal{X}$ | $\mathcal{X} \to \mathcal{Y}$ | $\mathcal{Y} \to \mathcal{X}$ |
| pix2pix | 9.39±2.3 | 7.27±1.5 | 9.60±3.0 | **8.50 ± 2.4** |
| BicycleGAN - Skip | 17.30±12.7 | 5.69±0.9 | 11.01±4.5 | 11.20±5.1 |
| BicycleGAN - No Skip | 10.61±2.2 | 6.63±1.0 | 14.19± 3.7 | 14.51±4.0 |
| Ours | **8.30±2.5** | **4.66 ± 1.1** | **8.28±2.9** | 9.45±4.5 |

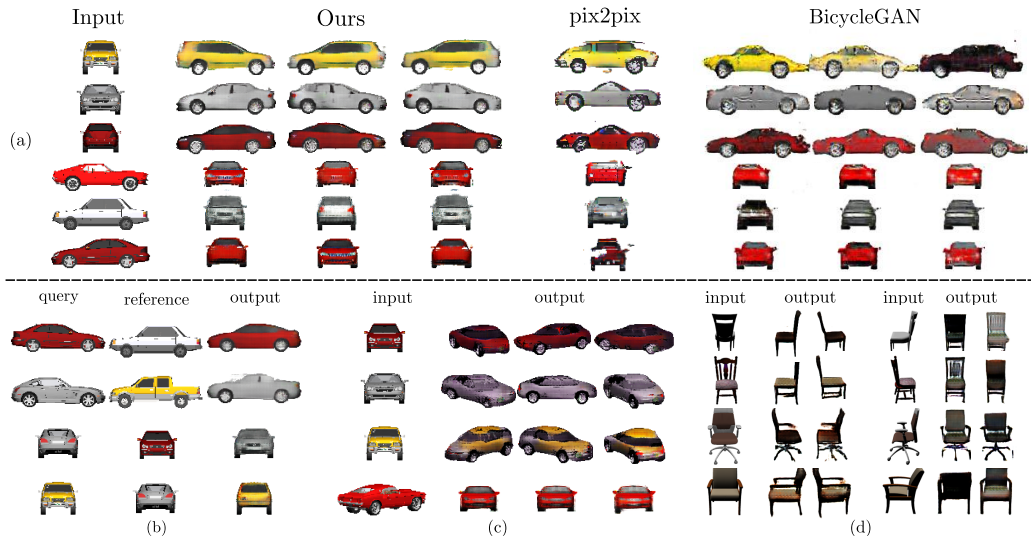

Figure 5: (a) Generated samples for the 3D car dataset [38] by our method, pix2pix [23], and BicycleGAN [50]. (b) Car analogies. (c) Samples when $\mathcal{Y}$ is all viewpoints except frontal. (d) Our generated 3D chairs [3].

many modes of the output distribution. Moreover, outputs are uni-modal when necessary (last row), as only domain-exclusive information is varied during generation and $\mathcal{X}$ only has one viewpoint.

**3D chair models.** Similarly to the cars dataset, [3] offers rendered images of 1393 CAD chair models from different viewpoints. We arrange $\mathcal{X}$ and $\mathcal{Y}$ as before, and train the model with 5,372 images from 1,343 chairs, leaving 50 chairs for test. Figure 5d shows examples of the chairs generated by our model. In this case, the samples also effectively cover both modes of the target distribution. Quantitatively (table 2), we achieve the best results in one direction but worse than pix2pix for the other. This could be due to the fact that viewpoint changes in this dataset are less extreme than with cars, as the aspect-ratios of frontal and profile views are quite similar.

## 4.3 Ablation study

We tried removing some network components and observed the effect on the model (table 3). We measure performance as the ability to create visual analogies, which guarantees a minimum of disentanglement. We create or select the ground-truth target analogy (e.g. digit of the query with the color of the reference) and compute the distance with the output of our model. We can see how some components such as the cross-domain autoencoders are crucial for this task, since when removed (No auto.) the performance decreases significantly. The performance drop when using normal autoencoders (Normal auto.) can be even greater. We attribute this to the information shortcut they introduce, which allows the model to ignore the exclusive information and makes the output of the visual analogy the reconstructed input. This effect is less noticeable for cars and chairs, as the domain modes are perceptually closer (e.g. same aspect ratio).

Our model manages to create visual analogies without some components, but it is negatively affected in other tasks (e.g. diverse sample generation) as well as training stability. For example, removing the GRL may increase the amount of shared information in the exclusive representation $E$. We measure this by performing intra-domain retrieval on each MNIST domain using only $E$. The average increase in recall (i.e. more shared information) when removing the GRL is only 0.7% for the current setting ($E$=1x8), but it grows with $E$'s size: 8% for 1x128 and 25% for 8x8x256. Therefore, the GRL is beneficial for keeping shared information out of the exclusive representation for particular settings, but its effect is more limited in the current configuration and thus it could be safely removed.

Table 3: Ablation study based on performance of visual analogies, measured as the distance to the ground-truth. We use Euclidean distance ($\times 10^{-2}$) for MNIST and LPIPS ($\times 100$) for Cars and Chairs.

| Dataset | Full | No auto. | Normal auto. | No GRL | No noise | No $\mathcal{L}_S$ | No $\mathcal{L}_{\text{recon}}$ |
|---|---|---|---|---|---|---|---|
| MNIST-CB | $13.0 \pm 3.2$ | $35.1\pm13.2$ | $40.0 \pm 11.5$ | $11.4 \pm 3.2$ | $13.0 \pm 3.7$ | $16.6 \pm 5.9$ | $18.9\pm6.4$ |
| MNIST-CD | $10.2 \pm 2.6$ | $15.0 \pm 4.6$ | $16.8 \pm 5.04$ | $11.5 \pm 3.12$ | $11.9 \pm 3.0$ | $12.4 \pm 3.7$ | $12.9 \pm 3.1$ |
| Cars-P | $8.9 \pm 2.8$ | $18.3 \pm 2.4$ | $9.3 \pm 1.4$ | $8.7 \pm 1.9$ | $8.7\pm2.8$ | $17.5 \pm 0.8$ | $10.6 \pm 1.9$ |
| Cars-F/B | $5.4 \pm 1.6$ | $5.7 \pm 1.9$ | $5.6 \pm1.5$ | $5.6 \pm 1.7$ | $5.6 \pm 1.6$ | $18.0 \pm 2.5$ | $6.0 \pm 1.9$ |
| Chairs-P | $11.4 \pm 2.4$ | $14.3\pm2.9$ | $14.3 \pm 3.1$ | $11.3 \pm 2.5$ | $12.0 \pm 1.8$ | $12.0 \pm 2.2$ | $15.4 \pm 1.5$ |
| Chairs-F/B | $12.1 \pm 2.8$ | $10.6 \pm 2.8$ | $9.9 \pm 3.2$ | $12.9 \pm 3.9$ | $10.1 \pm 3.2$ | $13.9 \pm 3.7$ | $10.2 \pm 3.4$ |

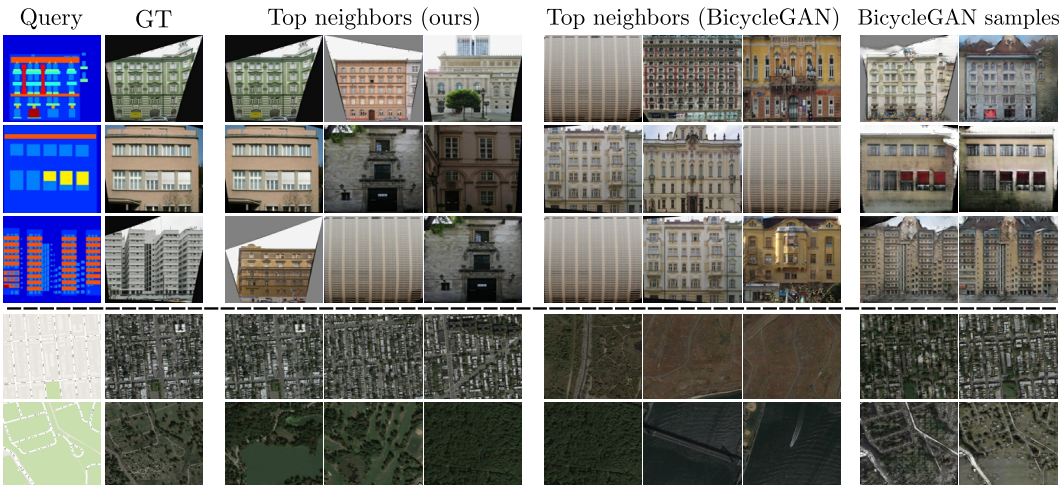

Figure 6: Retrieval results on Facades [45] and Maps [23] with our method and BicycleGAN [50]. We show the top-3 neighbors using each approach and BicycleGAN's generated samples. Each last row presents failure cases.

## 4.4 Cross-domain retrieval on realistic datasets

We have so far demonstrated representation disentanglement only on synthetic datasets. We explore here the applicability of our method to other datasets by presenting cross-domain retrieval experiments on two realistic datasets: maps $\leftrightarrow$ satellite images [23] and labels $\leftrightarrow$ facades [45], and comparing with BicycleGAN [50]. Our method enables cross-domain retrieval through the shared features (sec. 4.1). On the other hand, BicycleGAN does not provide an obvious way to tackle the retrieval task, as their internal features (both $E$'s output and $G$'s bottleneck [50]) are not disentangled and thus contain domain-specific information. One possible retrieval approach with BicycleGAN is: (i) translate the input image to the target domain (e.g. 10 random samples) and (ii) retrieve the most similar images to any of the translated samples. Table 1 (right) presents results using the provided train/test splits (100 test images for Facades and 1K for Maps) and the available pre-trained BicycleGAN models [50]. We measure performance as the percentage of cases for which the top retrieved image is the corresponding paired image. The excellent results of our model demonstrate representation disentanglement also on realistic datasets, which is an advantage over BicycleGAN given its poor results. Figure 6 shows qualitative results. Our method is able to abstract the patterns exhibited for both domains (e.g. window size, street directions) in the shared representation, which results in retrieved images that follow those patterns. Retrieval with BicycleGAN, however, generally fails at retrieving the corresponding image using the generated samples.

## 5 Conclusions

We have presented the concept of cross-domain disentanglement and proposed a model to solve it. Our model effectively disentangles the representation into a part shared across domains and two parts exclusive to each domain. We applied this to multiple tasks such as diverse sample generation, cross-domain retrieval, domain-specific image transfer and interpolation. We have tested on several datasets of different complexity, both synthetic and of real images. We also introduced the many-to-many image translation setting and paved the way to overcome some limitations of current approaches through the use of a disentangled representation.

**Acknowledgments**    We acknowledge the Spanish project TIN2016-79717-R and the CHISTERA project M2CR (PCIN2015-251).

## Footnotes

[1] We also investigated constraining $\left\| S^{\mathcal{X}} \right\| = \left\| S^{\mathcal{Y}} \right\| = 1$ but found this to be less stable.

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
