[Supplementary Material]

# Image-to-image translation for cross-domain disentanglement - Supplementary material

## A    Network architecture and hyperparameters

In this section, we disclose the network architecture and implementation details used for each experiment. We train our models with the recommended weight loss values of pix2pix, i.e. $w_{\text{GAN}} = 1$ and $w_{\text{L1}} = 100$. We set the weight $w_{\text{Ex}}$ of the exclusive decoder loss to 0.1. We found experimentally that a lower value in this case achieves a good representation disentanglement without excessively altering the quality of the generated images. The weighting parameter of the Gradient Reversal Layer, $\lambda$, is fixed to 1. We train with Adam and a fixed learning rate of 0.0002.

**Generator encoder - common:**   5 convolutional layers of size 4x4 and stride 2 with 64, 128, 256, and 512 filters, LeakyReLU of 0.2, and batch normalization.

**Generator encoder - shared:**   1 convolutional layer of size 4x4, stride 2, and 512 filters.

**Generator encoder - shared:**   1 fully connected layer with 8 output units.

**Generator decoder:**   5 convolutional layers of size 4x4 and stride 1/2 with 512, 256, 128, 64, and 3 filters, ReLU, and batch normalization. Dropout with probability 0.5 on the first 3 layers, only at training time.

**Exclusive generator decoder:**   same architecture as generator decoder, but with a Gradient Reversal Layer at its input.

**Discriminator WGAN-GP:**   3 convolutional layers of size 4x4 and stride 2 with 64, 128, and 256 channels, LeakyReLU of 0.2, and 1 fully connected layer with 1 output channel and without no-linearity.

**Discriminator GAN:**   3 convolutional layers of size 4x4 and stride 2 with 64, 128, and 256 channels, LeakyReLU of 0.2, batch normalization, and 1 last convolutional layer with 1 output channel and sigmoid as no-linearity.

**Input resolution:**   $256 \times 256 \times 3$, input image resized when necessary.

**Epochs:**

- MNIST-CD/CB: 15
- 3D cars: 900
- 3D chairs: 75
- Facades: 200
- Maps: 200