[Reviews · NeurIPS 2018]

Reviewer 1



In overall, I think this paper proposes a nice framework to learn the disentangled features. The authors addressed my questions in the rebuttal. And I suggest the author add the original BicycleGAN (with skip connections) results to the final version if the paper is accepted. Besides, I also suggest the author soften the claim about GRL loss or explain when the GRL loss can improve the performance and when not. My final rating is accept. ======================================== This paper proposes a framework to disentangle the domain-exclusive factor and domain-shared factors for paired image-to-image translation task. Pros: * The proposed cross-domain autoencoder pipeline is an interesting way to implement the domain-shared constraint. It seems a good alternative to the weight-sharing technique which is used in previous image-to-image translation and domain adaptation works. * Empirical experiments show that the proposed method can learn the disentangled features successfully. Cons: * In the abstract line 13, it is mentioned that the proposed method can perform cross-domain retrieval without the need of labeled. I am confused about this setting because the proposed method does need paired data for training, right? Besides, in line 269, it is mentioned that “Moreover, we do not need image labels, as opposed to specialized approaches such as [13].” However, [13] does not need paired data. So, what does the image labels mean here? * In line 28, it is mentioned that “In spite of the current popularity of these models, the learned representation (the output of the encoder), has not been studied.” I do not agree with this strong statement, since there are many related works on disentangled representation learning just not for image translation specifically, e.g. 1) Disentangled Representation Learning GAN for Pose-Invariant Face Recognition, in CVPR’17; 2) Disentangled person image generation, in CVPR’18. * In line 298, the authors remove the skip connections in [5,12] in order to do a fair comparison. I do not think it is a fair setting. Because the skip connections are very important for the architecture used in [5,12], while the proposed method cannot utilize the skip connections. Given that this paper is about the disentangled image-to-image translation, it is suggested that some recent work on disentangled representation learning can be included in the related work, e.g., [a] K. Bousmalis, G. Trigeorgis, N. Silberman, D. Krishnan and D. Erhan. “Domain separation networks.” In NIPS, 2016. [b] L. Ma, Q. Sun, S. Georgoulis, L. Van Gool, B. Schiele and M. Fritz. “Disentangled person image generation.” In CVPR, 2018. [c] Y.C. Liu, Y.Y. Yeh, T.C. Fu, S.D. Wang, W.C. Chiu and Y.C. Frank Wang. “Detach and Adapt: Learning Cross-Domain Disentangled Deep Representation.” In CVPR, 2018.

Reviewer 2



Paper main ideas/claims: This work presents a technique for performing image-to-image translation based on Generative Adversarial Networks (GANs) and a novel auto-encoder approach, which is named cross-domain autoencoder. The approach relies on partitioning internal representations into shared components between the input and output domain (X and Y, respectively) and also the exclusive components to X and Y. The authors test their results on a variation of MNIST, a 3D CAD car and a 3D CAD chair dataset. Strengths: I found the paper to be extremely clear and well written. The architecture proposed is novel, simple, and has very good results on the three datasets studied when compared to sota models for image translation such as pix2pix and BicycleGAN. The generated samples for each domain are diverse and seemingly avoid mode collapse. The results on cross-domain and domain agnostic retrieval without the need for labeled data are very encouraging. Weaknesses: The main weakness of this paper is not testing in a more realistic dataset. Moreover, I found the ablation study to be a bit on the weak side. It would have been interesting to see the results for the 3D car and chair datasets as well, since they are the most challenging in terms of internal representation. For example,Table 2 indicates that for the MNIST-CB and MNIST-CD datasets, not having an autoencoder is better than having a standard one -- which seems a bit odd. Same comment holds for the domain-specific interpolation, cross-domain and domain agnostic retrieval results. Error estimates in the numerical results are missing. Finally, one potential limitation of the approach proposed is the reconstruction loss from equation (2), which might only work well for domains in which the 'exclusive part' can be modeled by a low dimensional vector -- since otherwise the amount of noise added to the inputs might be too large. ------------------------------------------------------------------------ Post rebuttal comments: I really appreciated that the authors addressed most of the concerns raised by the reviewers. For example, extending their ablation studies and testing on more realistic datasets: facades and maps. I keep my initial grade, and think the paper should be accepted.

Reviewer 3



After reading the rebuttal and other reviews I decided to raise my score to 6. However I strongly suggest the authors to soften the claim about GRL, since I do not think it has the claimed effect even in theory. For example, even if the exclusive code has a Gaussian distribution which does not contain information of another domain, the generator can still generate images of another domain from it just like an unconditional GAN. The gradient receive by the encoder does not have a clear efffect. =========================== This paper proposes a new method for image-to-image translation. The model consists of two autoencoders, whose latent code is decomposed into a shared component and a domain specific component. The model supports multimodla image translation, generating different output images given the same input. Pros: 1. The paper is well-written and easy to follow. 2. The idea of multimodal image translation by decomposing domain-shared and domain-specific representation is novel, although the authors might want to consider discussing the difference from some concurrent works [1,2]. Cons: 1. To me, the GRL design does not really make sense. The GRL encourages the encoder to produce embeddings such that the images generated by G^X_d are different from images in domain Y. The encoder could achieve this by arbitrarily shifting its output distribution, for example, adding a bias of 100. The encoded feature will still contain the cross-domain information, but the images decoded from G^X_d will be garbage. Also, Table 2 indeed shows GRL does not improve performance. Is there any theoretical or empirical results to support the use of GRL? 2. The experiments are all done on synthetic datasets. It would be better to extend it to real-world datasets. 3. The method assumes paired data is available, similar to BicycleGAN. However, the advantage over BicycleGAN is not very clear. In Table 1, the performance of the proposed method is similar to BicycleGAN on Chairs dataset, and slightly better on Cars dataset. It could be that the default parameters of BicycleGAN do not work very well on the new dataset. To me it's not very convincing to say the proposed model outperforms BicycleGAN. Evaluating the proposed method on the datasets BicycleGAN originally used (e.g., edges2shoes) would stronger the claim. [1] Huang, Xun, et al. "Multimodal Unsupervised Image-to-Image Translation." arXiv preprint arXiv:1804.04732 (2018). [2] Almahairi, Amjad, et al. "Augmented CycleGAN: Learning Many-to-Many Mappings from Unpaired Data.", ICML 2018